# Simultaneous Extraction of Bioactive Compounds from *Olea europaea* L. Leaves and Healthy Seed Oils Using Pressurized Propane

**DOI:** 10.3390/foods12050948

**Published:** 2023-02-23

**Authors:** Jonas Marcelo Jaski, Rayane Monique Sete da Cruz, Tatiana Colombo Pimentel, Natalia Stevanato, Camila da Silva, Carlos Eduardo Barão, Lucio Cardozo-Filho

**Affiliations:** 1Department of Agronomy, State University of Maringa (UEM), Av. Colombo, 5790, Maringa 87020-900, PR, Brazil; 2Federal Institute of Parana, Paranavai Campus, Av. Jose Felipe Tequinha, 1400, Paranavai 87703-536, PR, Brazil; 3Department of Chemical Engineering, State University of Maringa (UEM), Av. Colombo, 5790, Maringá 87020-900, PR, Brazil; 4Research Center, Centro Universitario Fundacao de Ensino Octavio Bastos (UNIFEOB), São Joao da Boa Vista 13874-149, SP, Brazil

**Keywords:** chia, sesame, phytosterols, lipid profile, pressurized fluid

## Abstract

Olive leaves (OL) are products of olive cultivation with a high commercial value because they contain valuable bioactive compounds. Chia and sesame seeds have a high functional value because of their attractive nutritional properties. When combined in the extraction process, the two products constitute a product of high quality. The use of pressurized propane in vegetable oil extraction is advantageous because it provides solvent-free oil. This study aimed to combine two high-quality products to obtain oils with a unique combination of attractive nutritional properties and high levels of bioactive compounds. The mass percentage yields of the OL extracts with chia and sesame oils were 23.4% and 24.8%, respectively. The fatty acid profiles of the pure oils and their respective OL-enriched oils were similar. There was an aggregation of the 35% and 32% (*v*/*v*) bioactive OL compounds in chia and sesame oils, respectively. OL oils exhibited superior antioxidant capacities. The induction times of the OL extracts with the sesame and chia oils increased by 73% and 4.4%, respectively. Incorporating OL active compounds in healthy edible vegetable oils using propane as a solvent promotes the reduction of lipid oxidation, improves the lipid profiles and health indices of the oils, and forms a product with attractive nutritional characteristics.

## 1. Introduction

Lipid oxidation in vegetable oils is a crucial factor governing their quality attributes and reduces their shelf life [1,2]. Synthetic antioxidants, such as butyl hydroxyanisole, butylated hydroxytoluene, tert-butylhydroquinone (TBHQ), and propyl gallate, have been used to overcome this undesirable effect and extend the shelf lives of the products [3]. However, specialized literature [4,5,6] shows that some synthetic antioxidants may have carcinogenic potential.

The use of natural antioxidants in vegetable oils is an alternative to avoid the toxic effects of synthetic antioxidants [7,8]. In addition, active compounds present in physiologically active natural antioxidants, such as phytosterols, are frequently used to develop functional foods [9,10]. In addition, recent research shows that vegetable oils can be used for the bio-accessibility of these active compounds [11].

In a recently published broad review, Gharby et al. [12] reported that extracts from aromatic and medicinal plants used as natural antioxidants can improve the nutritional values and oxidative stabilities of vegetable oils. Olive leaf (OL) extracts can be used to enrich vegetable oils, which consequently exhibit increased oxidative stability and active compound content. OL compounds demonstrate a high potential for use as natural antioxidants [8,12,13,14]. However, there is much to be explored, such as vegetable oils with a nutritionally attractive chemical composition. Because of the presence of valuable active compounds in OL, its extracts are being used to preserve and improve the quality of agroindustry products [15,16,17].

Vegetable oils constitute the main source of lipids in our diet. Chia (*Salvia hispanica* L.) and dark sesame (*Sesamum indicum* L.) oils have been studied because of the presence of bioactive compounds, such as essential fatty acids (FA), phytosterols, tocopherols, and other compounds that act as antioxidants [18,19,20]. Additionally, the consumption of phytosterols and tocopherols promotes other health benefits, such as the reduction of low-density lipoprotein (LDL) cholesterol and the prevention of chronic diseases [21].

Chia seeds contain approximately 30% oil and are consumed because of their high contents of phytosterols; tocopherols; and polyunsaturated FAs (PUFA), especially of the omega-3 class that is mainly responsible for preventing heart disease [20,22]. Sesame seeds contain approximately 48% oil, and dark seeds have less variation in contents than light seeds. The main unsaturated FAs (UFA) present in the seeds are oleic and linoleic acids [19,23]. 

Green extraction techniques, which are considered more efficient and less environmentally aggressive, are gradually replacing conventional techniques that harm consumer health owing to toxic organic solvent residues in the final products [24].

Among the green extraction techniques, the technology of extraction with pressurized fluids is attractive because it has high selectivity, the absence of light and oxygen, and uses mild temperatures that avoid the degradation of thermolabile compounds [25,26,27].

The use of n-propane as a pressurized fluid for vegetable oil extraction is notable. Once n-propane has a high solubility in vegetable oils, it requires a small amount of solvent and a shorter extraction time when compared with conventional organic solvents (hexane) or pressurized carbon dioxide [24,28,29]. 

Studies on the association of bioactive compounds present in OL with chia and sesame vegetable oils via extraction with pressurized n-propane, resulting in an increased lipid induction time (IT) and nutritional quality of the product, are scarce. Furthermore, durable and sustainable techniques to extract natural antioxidants are innovative ways to achieve a circular economy and meet the natural and healthy food needs of consumers. Thus, the selection of extraction methods is important for a proper enrichment process in functional food design [12,30].

In this context, this study was aimed at filling this gap by joining two raw materials of high added value to obtain a product with superior characteristics in a more sustainable way using green technology, free of solvent residues. The percentage lipid yield, FA profile, bioactive compound content, oxidative stability, and antioxidant activity, were determined to prove the beneficial effects of the active compounds on vegetable oils using statistical analysis techniques.

## 2. Methods

### 2.1. Samples

Sesame and chia seeds (500 g each) were purchased from a local market in Maringá, Paraná, Brazil. The chia and sesame varieties obtained were domesticated varieties of black chia and sesame, respectively. Seeds were comminuted to the smallest size (2.0 mm) using a manual coffee grinder. The OL of the arbequina, koroneiki, and arbosana varieties used in this study were obtained at high altitudes in the State of São Paulo (22°00′48.6″ S, 46°37′59.4″ W). A forced ventilation oven (Ethik Technology, Vargem Grande Paulista, Brazil, model 400/4ª, Vargem Grande Paulista, SP, Brazil) at 35 °C was used for 36 h to reduce the moisture content of OL from 60% to approximately 5%. Subsequently, the dry leaves were crushed in a rotary knife cutter (model SL-30, Solab, Piracicaba, SP, Brazil) with a granulometry of ≤2.0 mm and homogenized.

### 2.2. Extraction with Pressurized Propane 

Extractions using propane (Messer, Bad Soden am Taunus, Germany, purity 99.5%) were performed in duplicate using an experimental apparatus previously described by Trentini et al. [31]. 

Initially, extractions were performed with chia and sesame seeds using 20.0 g of seeds per extraction, which were added to an extraction vessel for complete filling (53.4 cm^3^). 

Extractions using seeds + OL were performed using a structured bed composed of OL and oilseeds (chia or sesame) in a 3:1 (*v*/*v*) proportion. The bottom of the extractor (165.2 cm^3^) was first loaded with 26.0 g of the leaves, followed by the sequential addition of 20.0 g of the seeds. When pressurized, propane first interacts with the seeds, extracting the oil. This acts as a co-solvent in this process, extracting the compounds of interest from OL. 

The temperature (60 °C), pressure (12 MPa), volumetric flow rate (1.5 mL·min^−1^), and extraction time (60 min) were defined based on previous studies [19,20]. These operating conditions resulted in the highest oil yield of the seeds used. A syringe pump (Teledyne ISCO 500D) fed pressurized n-propane into the jacketed extractor. Samples of the obtained oil were collected and weighed at intervals of 5 min in an amber glass bottle. The mass percentage yield was calculated from the weights of the oil samples.

### 2.3. Fatty Acids and Bioactive Compounds

A gas chromatograph coupled with a mass spectrometer (Shimadzu, Kyoto, Japan, GC-MS QP2010 SE) equipped with an automatic injector (AOC-20i) was used to determine the bioactive compounds and AGs present in the samples. Jaski et al. [32] reported the analytical protocol and analysis conditions.

The FA profile was determined after saponification of the oil and derivatization with boron trifluoride (14% in methanol, Sigma-Aldrich, >99% purity, St. Louis, MO, USA) as described by Gonzalez et al. [33]. The total area of the FAs that flowed into the capillary column was used to determine the relative percentage of each FA.

A broad-based methodology was used for the identification and quantification of phytosterols and tocopherols [34] using a capillary column SH-Rtx-5MS™ (Shimadzu, 30 m × 0.25 mm × 0.25 μm). The samples were derived from N,O-bis (trimethylsilyl) trifluoroacetamide trimethylchlorosilane (BSTFA/TMCS, Sigma-Aldrich) for 30 min at 60 °C. The 5α-cholestane standard (Sigma-Aldrich, >99% purity) was added to the samples, which were then diluted with heptane (anhydrol, >99% purity). The column heating ramp used during the analysis was reported by Santos et al. [35], Trentini et al. [31], and Cuco et al. [36].

The quality of the lipid fraction of oils for human consumption can be evaluated using indices that indicate health, such as the atherogenic index (AI), thrombogenic index (TI), desired FAs (DFA), and hypercholesterolemic saturated FAs (HSFA). The health indicator indices were obtained using Equations (1) [37], (2) [38], (3), and (4) [39,40].
(1)AI=(12:0+4×4:0+16:0)/[MUFA+(n−6)+(n−3)
(2)$$\begin{array}{cc} I=(12:0+14 & :0\\ & +18:0)\\ & /[0.5\times \sum^{\;}\mathrm{MUFA}+0.5*\left(n-6\right)+3*\left(n-3\right)\\ & +\left(n-6\right)/\left(n-3\right)] \end{array}$$
(3)DFA=MUFA+PUFA+C18:0
(4)HSFA=12:0+14:0+16:0
where 12:0, 14:0, and 16:0 correspond to lauric, myristic, and palmitic acid, respectively. ‘MUFA’ is the sum of the monounsaturated FAs (MUFA). ‘PUFA’ is the sum of the PUFA, n-3 is the sum of the omega-3 PUFA, and n-6 is the sum of the omega-6 PUFA.

For phytosterol and tocopherol analysis, the oils were derivatized with BSTFA/TMCS (Sigma-Aldrich) for 30 min at 60 °C, and 5α-cholestane standard (Sigma-Aldrich, >99% purity) was added to the derivatized samples. 

To calculate the percentage of active compound aggregation of OL in vegetable oils, the number of active compounds in the OL extract was 100%. The percentage of active compounds contained in the oil prior to the addition of the OL extract was discounted, and the percentage of aggregation in the oil was mathematically obtained.

### 2.4. Antioxidant Activity 

Antioxidant activity was analyzed via the 1,1-diphenyl-2-picrylhydrazine (DPPH) free radical sequestration method [41,42] using three duplicates of each sample. 

Ethanol (Anhydrol, 99.5% purity) was used to dilute the oil samples, oil incorporated with the OL extract, and the OL extract to an initial concentration of 2000 µg·mL^−1^ each. Samples of the solution (250, 500, 750, 1000, 1500, and 2000 μL) were transferred to test tubes containing 2.0 mL of a DPPH methanolic solution (Anhydrol, 99.5% purity) (Sigma-Aldrich, purity ≥ 90%). After incubation for 30 min in the dark, the absorbance was measured using a spectrophotometer (Hach/DR 2800) at 517 nm. Methanol (Panreac, 99.9% purity) was used as the reference. The DPPH antioxidant activity was calculated using the following Equation (5):(5)$$A_{DPPH}\left(\%\right)=\left(\frac{A_{DPPH}-\left(A-A_{B}\right)}{A_{DPPH}}\right)\times 100$$
where *A_DPPH_* is the absorbance of the *DPPH* solution, and *A* and *A_B_* are the absorbances of the samples and blank, respectively. 

The sample concentration capable of reducing 50% of *DPPH* (EC_50_) was calculated from the linear equation of percentage antioxidant activity versus sample concentration (μg mL^−1^).

### 2.5. Oxidative Stability

A Metrohm 873 Biodiesel Rancimat oxidation stability analyzer (Metrohm, Herisau, Switzerland) was used to determine the oxidative stabilities of the oils. The sample (3.0 g) was exposed to airflow of 10 L·h^−1^ at a constant temperature of 120 °C in a reaction vessel. The IT was automatically determined from the second derivative of the conductivity curve using StabNet Software (version 1.1) [43].

### 2.6. Statistical Analysis

The extraction and characterization experiments were performed in duplicate, and the analysis was performed using RSTUDIO (RSTUDIO, Inc., Boston, MA) and SAS software (SAS Institute Inc., Cary, NC, USA). The student’s *t*-test was used to compare mean values between pairs of oils (with and without OL), and Tukey’s test was conducted to compare the means between all treatments, with both using *p* < 0.05.

Principal component analysis (PCA) was performed using the FA profile, bioactive compounds, antioxidant activity (DPPH), yield, and IT. A matrix consisting of 4 rows (oil types) and 24 columns was used. PCA and Pearson’s correlation were performed using XLSTAT^®^ 2021.4.1 software (Addinsoft™, Paris, France).

## 3. Results and Discussion

### 3.1. Mass Percentage Yield

Table 1 shows the percentage yields (*w*/*w*) of oils obtained from seeds with and without the addition of OL, as well as the OL extract. The percentage oil yield in the different species evaluated ranged from 23–31%, with sesame providing the highest yield.

The percentage yields agree with those reported in the literature under the same experimental conditions used in this study, only using the seeds as the raw material and propane as the solvent. Corso et al. [19] and Zanqui et al. [20] observed 34% and 27% yields in sesame and chia seeds, respectively. These studies showed slightly higher extraction yields using hexane and ethers than those obtained in this study; however, they did not use environmentally friendly solvents. The slight differences found between the yields of oils and natural extracts depend on several factors, including not only the solvent and experimental conditions but also the characteristics of the raw material. Plant cultivation in different soil types, climatic conditions, stress conditions, and times of harvests can influence oil yield [44,45,46].

In all cases, oil extraction with seeds + OL showed lower yields than those performed with seeds alone. This behavior can be explained by the physical impediments of the shredded leaves favoring preferred pathways inside the extraction vessel, low oil content in the composition of OL, and low solubility of OL components in propane. Results such as the extractions performed on pumpkin skin with pumpkin seeds were reported by Cuco et al. [26]. 

Pressurized propane has been reported as an efficient solvent for oil extraction with a short extraction time and high yield [47]. The present results show that propane is also efficient for the extraction of chia and sesame oils incorporated with active compounds; however, the proportion of raw materials in the extraction vessel should be further explored in studies so that the yield is unaffected and the active compounds are used efficiently.

The kinetic curves for the extraction of vegetable oils from different seeds are shown in Figure 1.

The curves show that using the leaves along with the seeds in the extraction vessel did not achieve kinetic stabilization. This behavior may be due to the low solubility of the OL components in the solvent and blockage in the passage of oil due to the shredded leaves. Thus, the oil took more time to reach the collection container, which did not affect the final extraction content. According to the literature, even if the kinetic curves do not reach stabilization, a major portion of the oil contained in the seeds is extracted during the initial 60 min [19,20].

### 3.2. Fatty Acids

Table 2 shows the relative FA compositions of all oils obtained.

Chia oil contained approximately 72% PUFA, especially α-linolenic acid (n-3), and sesame oil contained high amounts of n-9 (oleic) and n-6 (linoleic) FA. Simultaneous extraction of the raw materials (leaves and seeds) did not change the lipid profile of the oil and increased the quality of the product. The lipid profiles of oils obtained using propane are of high quality and are supported by the literature, with little variation in quantification [19,20]. Factors such as varying conditions in plant cultivation, growing region, and material collection time affect secondary metabolite production, resulting in slight variations in the lipid profiles [44,45,46]. 

The AI, TI, DFA, and HSFA were calculated to verify the quality of the lipid fractions of the oils for human ingestion. Low AI and TI values indicate a great potential to prevent the onset of coronary heart diseases [48]. A high content of DFA is related to an increase in the nutritional quality of edible oils, and HSFA is related to increased hypercholesterolemia [40].

The AI and TI indices were low in all oils obtained, with chia oil having the lowest TI. AI and TI were not statistically different (*p* > 0.05) between oils with and without the OL extract. According to Sperry et al. [38], a high-quality lipid fraction has low AI and TI, indicating that these oils may inhibit platelet aggregation and prevent the onset of coronary disease. 

DFA is considered to have neutral or cholesterol-reducing effects on food [39]. Chia oil has an advantage over sesame oil in that it contains almost twice the percentage of DFA. This is because chia oil has a higher percentage of PUFA and a lower percentage of SFA than sesame oil. Chia oil showed higher DFA than chia + OL because the OL extract contains small amounts of SFA [32], influencing the DFA. For sesame oil, there was no change in DFA when incorporated into the OL extract, demonstrating that different raw materials can exhibit different behaviors during simultaneous extraction.

### 3.3. Bioactive Compounds

Table 3 shows the quantification of bioactive compounds present in the vegetable oils and OL extracts.

Eight bioactive compounds were identified in the analyzed oils, the main compounds of which were α-tocopherol, campesterol, stigmasterol, and β-sitosterol. Table 3 shows the statistical difference for each seed pair (with and without OL extract), where different letters in the same row indicate a significant difference between the student’s *t*-tests (*p* < 0.05). 

The OL extract contained high levels of active compounds, mainly β-sitosterol, α-tocopherol, and 1-octacosanol. This finding is indispensable for the increase in these compounds in vegetable oils. The high concentration of active compounds in the OL extract was responsible for the increase in these compounds in vegetable oils after structured bed extraction.

α-Tocopherol was not found in any oil and increased in all oils obtained with OL. Tocopherols are potent antioxidants, and their antioxidant properties are attributed to their aromatic ring hydroxyl groups, which donate hydrogen to neutralize free radicals or reactive oxygen species (ROS) [49]. α-Tocopherol also fights cancer and provides protection against bone, cardiovascular, eye, and neurological diseases [49,50]. Therefore, the increase in α-tocopherol by OL extract content in vegetable oils provides multiple health benefits. 

In this study, the phytosterols found were 1-octacosanol, campesterol, β-sitosterol, stigmasterol, 1-triacontanol, and pregn-5-en-3-ol, which were increased by the OL extract. Table 4 shows the percentage aggregation of the main OL bioactive compounds in the vegetable oils. The oils enriched with phytosterols can be incorporated into functional foods, favoring the reduction of LDL cholesterol.

The aggregate contents of α-tocopherol were 8.5% and 6.3% in chia and sesame oils, respectively. β-Sitosterol was incorporated into oils with OL, with 6.5% and 5.4% in chia and sesame oils, respectively. The final products, which are lipid sources of high nutritional quality, were successfully enhanced with active compounds. The highlight was chia oil, with approximately 35% aggregation of the bioactive compounds. In addition, sesame oil exhibited an interesting aggregation percentage of 31.3%. The variation in the amount of active compound aggregation depends on the composition and intrinsic characteristics of each oil that acts as a co-solvent in the extraction process when used in a structured bed [36]. 

Vegetable oils can be considered efficient carriers for the bio-accessibility of these bioactive compounds. Bio-accessibility depends on the characteristics of an oil and its components. [11,51]. In the case of chia oil, which contains high concentrations of PUFA, it is more effective to incorporate the active compounds in OL. However, sesame oil was also effective in incorporating these active compounds. Therefore, the proposed methodology can be used as an alternative to determining the bioavailability of active compounds.

### 3.4. Antioxidant Activity

Table 5 presents the antioxidant activities of pure vegetable oils and oils containing the OL extract.

Pure vegetable oils had high values of EC_50_, indicating little resistance to oxidation and low antioxidant activity. The use of OL in the extraction vessel provided vegetable oils with high antioxidant activities, which was corroborated by the low EC_50_ values. Statistical analysis showed that all oils containing the OL extract had higher antioxidant activity than their counterparts devoid of the extract (*p* < 0.05). The highest antioxidant activity was related to the high content of bioactive compounds in oils obtained from the OL extract (Table 3). 

Comparing all samples (capital letters in the column), the statistical analysis showed that the OL extract had the highest antioxidant activity, followed by sesame oil + OL, the oil with the highest antioxidant activity, and pure sesame oil and chia oil + OL.

Similar results were obtained by Cuco et al. [26] for pumpkin seed oil extracted in conjunction with pumpkin peels. The authors reported that EC_50_ decreased from 817 μg·mL^−1^ in seed oil to 554 μg·mL^−1^ in seed oil + peel.

It is believed that the higher antioxidant activity was due to the presence of α-tocopherol, since tocopherols have antioxidant activity in lipid-based systems well described in the literature [52,53]. However, other compounds, such as squalenes, pigments, and sterols, can also increase the antioxidant activities of vegetable oils [54].

The antioxidant activity of the extract used as a natural antioxidant is related to the possibility of using this material for food enrichment [7]. Vegetable oils, particularly when used as co-solvents for the extraction of active compounds with antioxidant action, can provide a novel alternative approach for industrial expansion in the food industry [55] because they can select high-affinity compounds, providing efficient incorporation of compounds in the final product.

### 3.5. Oxidative Stability

Table 5 shows the IT of the different oils obtained with and without the OL extract.

Chia oils are quickly oxidized owing to their high PUFA composition [56]. There was no significant variation in oil IT with or without OL extract (0.46 h), and the present IT was like that obtained by Zanqui et al. [20]. For sesame oil, samples with the OL extract showed higher resistance to oxidation than their respective counterparts without the OL extract.

Corso et al. [19] extracted sesame oil with propane and reported an IT of 3.18 h, which is like that obtained in this research (3.12 h). According to the statistical analysis (*p* < 0.05), sesame oil incorporated with the OL extract showed a higher IT (5.4 h) than sesame oil without the OL extract. The IT of sesame oil with the OL extract increased by 159%. The IT of sesame oil treated with OL extract increased by 159%. The 2.27 h increase in IT resembles the IT increase found for the synthetic antioxidant TBHQ (3.14 h) used in sunflower oil [57].

Studies suggest the development of blends of chia oil with other oils, such as sesame oil, to provide lipid matrices rich in ω-3 FA that are more resistant to oxidation [58,59]. This may be a new subject of study for the simultaneous extraction of the raw materials of chia, sesame, and OL, in different proportions.

Given the significant increase in the oxidative stability of sesame oil extracted with OL, it is believed that the mixture of incorporated active compounds, such as α-tocopherol, and the other active compounds present in OL and sesame oil (rich in active compounds) has a synergistic effect on oxidative protection. Recent research has shown that antioxidant compounds such as tocopherols are efficient in increasing the storage stabilities of vegetable oils and that a mixture of antioxidants can have a synergistic effect in protecting against lipid oxidation [60].

### 3.6. Principal Component Analysis 

The first two principal components (PC1 and PC2) explained 94.72% of the total variance in the data (PC1: 62.83%; PC2: 31.89%) (Figure 2). PC1 separated the oils based on seed type, with sesame and chia oil on the right side of the axis and chia oil on the left. Sesame oil was characterized by longer IT and high concentrations of palmitic, palmitoleic, stearic, oleic, linoleic, arachidic, gondoic, and behenic acids, as well as fagarol, campesterol, stigmasterol, and pregn-5-en-3-ol. In contrast, chia oil was characterized by high concentrations of lauric, myristic, linolenic, and lignoceric acids, and low antioxidant activity. PC2 separated the oils based on OL, regardless of seed type. Thus, oils with OL were above the axis, and those without OL were below the axis. The inclusion of OL increased the content of 1-octacosanol, γ-tocopherol, α-tocopherol, 1-triacontanol, and β-sitosterol; and decreased the oil yield.

Appendix A presents the Pearson correlations for the evaluated attributes. Yield was inversely correlated with 1-octacosanol (*p* = 0.978), 1-triacontanol (*p* = 0.951), and β-sitosterol (*p* = 0.965). This result may be associated with the utilization of OL, which decreased the yield but resulted in increased incorporation of 1-octasanol (7.1–7.6%), triacontanol (7.6–9.1%), and β-sitosterol (5.4–6.5%) (Table 4). Simultaneously, high antioxidant activity was correlated with high levels of stigmasterol (*p* = 0.994). In fact, stigmasterol has received much attention owing to its antioxidant properties, which inhibit oxidation and prolong the shelf life of oils [61]. Finally, IT was inversely correlated with pregn-5-en-3-ol (*p* = 0.953). This compound has been reported as an important antioxidant [62]. 

## 4. Conclusions

Enhancing the content of bioactive compounds present in edible oils was possible by using propane pressurized as a solvent on a structured bed with OL and oilseeds. The oxidative stability values were compatible with those found in conventional techniques for adding antioxidants to oils. Statistical analyses showed that all oil samples extracted with OL had high levels of phytosterols and tocopherols, which are responsible for the increase in antioxidant capacity and oxidative stability. Lipid profiles and health indices were maintained in all edible oils studied. There was a remarkable increase in the IT of sesame oil with the use of OL on a structured bed. The elimination of post-processing steps, small amount of solvent, short extraction time, and high-quality oil obtained, make the simultaneous extraction process attractive. The poor bioavailability of bioactive compounds in plant matrices poses a challenge to process scalability. The proposed methodology has the potential to overcome this limitation and increase the bio-accessibility of active compounds. The proposed methodology aggregates chemically bioactive compounds in different vegetable oils without toxic solvents by combining high-quality oilseeds and OL. The structured bed extraction process was scalable and environmentally friendly.

## Figures and Tables

**Figure 1 foods-12-00948-f001:**
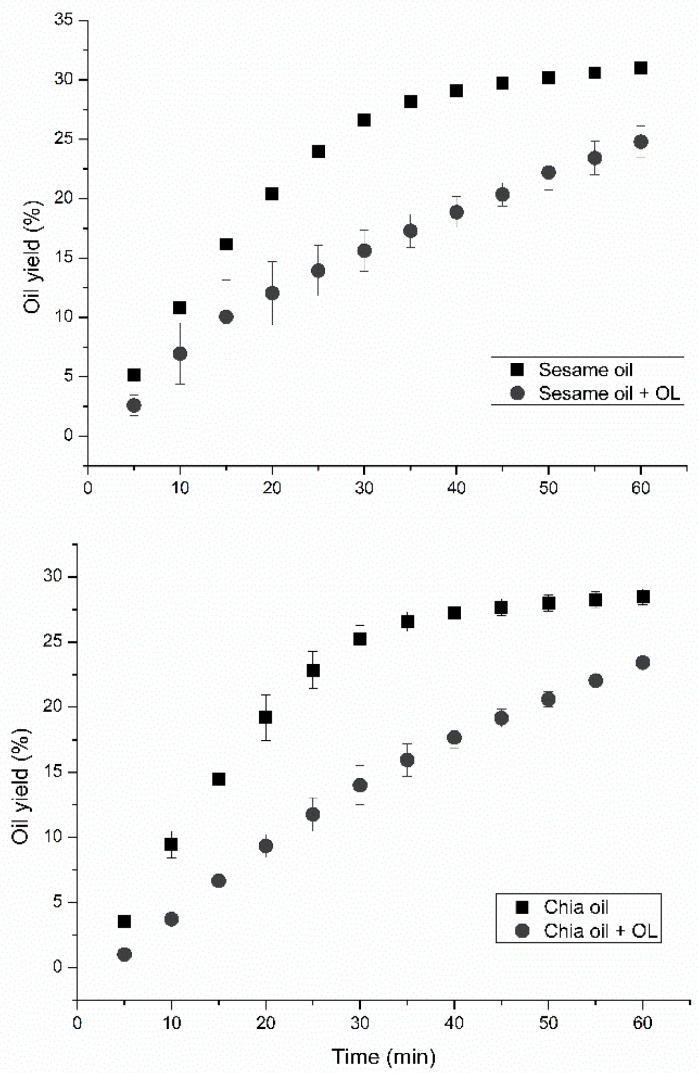
Kinetic curves of extraction of chia and sesame oils with and without the presence of OL.

**Figure 2 foods-12-00948-f002:**
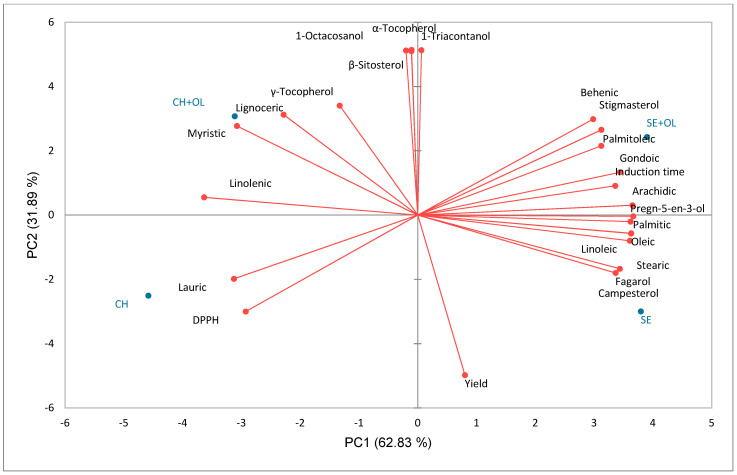
Principal component analysis (PCA) of oils. CH (chia oil), CH + OL (chia oil + OL extract), SE (sesame oil), SE + OL (sesame oil + OL extract).

**Table 1 foods-12-00948-t001:** Mass percentage yield of sesame and chia oils obtained with and without OL.

Sample	Yield (%)
Chia	28.5 ± 0.6 ^aA^
Chia + OL	23.4 ± 0.4 ^bB^
Sesame	31.0 ± 0.3 ^aA^
Sesame + OL	24.8 ± 1.3 ^bB^
OL	0.8 ± 0.2 ^E^

Mean ± standard deviation of duplicates. The lower-case letters in the column indicate a significant difference between oil yields extracted with and without OL by the t-student mean test (*p* < 0.05). Different capital letters in the column indicate a significant difference between oil production of different seeds by Tukey test (*p* < 0.05). OL: olive leaves.

**Table 2 foods-12-00948-t002:** Fatty acids and health indices of sesame and chia oils extracted with and without the presence of OL.

FA (Area%)	Chia	Chia + OL	Sesame	Sesame + OL
Lauric (12:0)	0.13 ^a^ ± 0.01	0.05 ^b^ ± 0.01	0.03 ^a^ ± 0.01	0.02 ^a^ ± 0.01
Myristic (14:0)	0.10 ^a^ ± 0.01	0.12 ^a^ ± 0.01	0.03 ^a^ ± 0.01	0.07 ^a^ ± 0.01
Palmitic (16:0)	9.30 ^b^ ± 0.04	9.59 ^a^ ± 0.01	11.02 ^a^ ± 0.03	11.02 ^a^ ± 0.01
Palmitoleic (16:1n-7)	0.09 ^a^ ± 0.01	0.13 ^a^ ± 0.01	0.14 ^a^ ± 0.01	0.15 ª ± 0.01
Stearic (18:0)	5.79 ^a^ ± 0.01	5.67 ^b^ ± 0.01	8.07 ^a^ ± 0.09	7.95 ^a^ ± 0.03
Oleic (18:1n-9)	9.52 ^a^ ± 0.04	9.49 ^a^ ± 0.03	40.37 ^a^ ± 0.06	39.94 ^b^ ± 0.08
Linoleic (18:2n-6)	20.95 ^a^ ± 0.07	20.74 ^a^ ± 0.04	38.20 ^a^ ± 0.13	38.11 ^a^ ± 0.11
α-Linolenic (18:3n-3)	51.58 ^a^ ± 0.05	51.15 ^a^ ± 0.05	0.42 ^b^ ± 0.01	0.87 ^a^ ± 0.01
Arachidic (20:0)	0.54 ^b^ ± 0.01	0.62 ^a^ ± 0.01	0.99 ^a^ ± 0.03	1.05 ^a^ ± 0.01
Gondoic (20:1n-9)	0.14 ^a^ ± 0.08	0.22 ^a^ ± 0.01	0.29 ^a^ ± 0.01	0.29 ^a^ ± 0.01
Behenic (22:0)	0.16 ^b^ ± 0.01	0.22 ^a^ ± 0.01	0.3 ^a^ ± 0.02	0.28 ^a^ ± 0.01
Lignoceric (24:0)	0.22 ^b^ ± 0.01	0.50 ^a^ ± 0.01	0.15 ^a^ ± 0.01	0.17 ^a^ ± 0.01
SFA	16.24 ^a^ ± 0.06	16.78 ^a^ ± 0.03	20.51 ^a^ ± 0.14	20.56 ^a^ ± 0.05
MUFA	10.95 ^a^ ± 0.13	11.07 ^a^ ± 0.05	40.87 ^a^ ± 0.08	40.45 ^a^ ± 0.11
PUFA	72.54 ^a^ ± 0.19	71.89 ^b^ ± 0.10	38.62 ^a^ ± 0.19	38.98 ^a^ ± 0.16
AI	0.12 ^a^ ± 0.01	0.12 ^a^ ± 0.01	0.14 ^a^ ± 0.01	0.14 ^a^ ± 0.01
TI	0.09 ^a^ ± 0.01	0.09 ^a^ ± 0.01	0.45 ^a^ ± 0.01	0.46 ^a^ ± 0.01
DFA	89.28 ^a^ ± 0.05	88.63 ^b^ ± 0.03	46.69 ^a^ ± 0.04	46.93 ^a^ ± 0.08
HSFA	9.52 ^b^ ± 0.05	9.77 ^a^ ± 0.01	11.07 ^a^ ± 0.03	11.11 ^a^ ± 0.01

Mean ± standard deviation of duplicates. Different letters in the same row indicate a significant difference between the t-student mean test (*p* < 0.05) for each seed pair (with OL and without OL extract). OL: olive leaves; SFA: sum of saturated fatty acids; MUFA: sum of monounsaturated fatty acids; PUFA: sum of polyunsaturated fatty acids; AI: Atherogenic index; TI: thrombogenic index; DFA: desired fatty acids; HSFA: hypercholesterolemic saturated fatty acids.

**Table 3 foods-12-00948-t003:** Bioactive compounds of sesame and chia oils obtained with and without the presence of OL and in the extract obtained only from leaves.

Bioactive Compounds (mg 100 g^−1^)	Sample				
	OL	CH	CH + OL	SE	SE + OL
α–tocopherol	631.4 ± 8.4	ND	53.6 ± 0.3	ND	40.0 ± 3.3
γ–tocopherol	ND	39.1 ^a^ ± 0.5	37.6 ^a^ ± 2.4	51.4 ^a^ ± 0.1	49.1 ^a^ ± 4.7
1–octacosanol	287.0 ± 4.7	3.3 ^b^ ± 0.1	23.8 ^a^ ± 1.5	ND	21.7 ± 2.0
Campesterol	ND	15.4 ^a^ ± 0.1	15.3 ^a^ ± 0.1	21.6 ^a^ ± 0.1	19.3 ^b^ ± 0.1
Stigmasterol	53.6 ± 0.8	11.8 ^a^ ± 0.4	14.1 ^a^ ± 1.0	14.9 ^a^ ± 0.7	17.2 ^a^ ± 0.8
1–triacontanol	147.3 ± 5.9	ND	13.4 ± 1.1	ND	11.2 ± 1.2
β–sitosterol	960.5 ± 22.7	82.0 ^b^ ± 0.9	143.9 ^a^ ± 0.8	77.7 ^b^ ± 1.8	129.3 ^a^ ± 0.8
Pregn-5-en-3-ol	ND	ND	ND	13.0 ^a^ ± 0.3	14.7 ^a^ ± 0.7

Mean ± standard deviation of duplicates. Different letters in the same row indicate a significant difference between the t-student mean test (*p* < 0.05) for each seed pair (with OL and without OL extract). ND: Not detected. OL: OL extract; CH: chia seed oil; CH + OL: chia seed oil + OL extract; SE: sesame seed oil; SE + OL: sesame seed oil + OL extract; OL: olive leaves.

**Table 4 foods-12-00948-t004:** Percentage of aggregation of bioactive compounds from OL in chia and sesame oils.

Aggregation (%)	Samples	
	Chia + OL	Sesame + OL
α—tocopherol	8.5	6.3
1—Octacosanol	7.1	7.6
Stigmasterol	4.3	4.4
1—triacontanol	9.1	7.6
β—sitosterol	6.5	5.4
Total	35.5	31.3

OL: olive leaves.

**Table 5 foods-12-00948-t005:** Antioxidant activities (EC_50_) of pure chia and sesame oils, chia and sesame oils extracted simultaneously with OL and pure OL extract, all obtained via pressurized propane. Induction time (h) of pure chia and sesame oils and the same oils extracted simultaneously with OL via pressurized propane.

Sample	EC_50_(μg mL^−1^)	Induction Time (h)
Chia	914.1 ± 41.3 ^aD^	0.45 ± 0.01 ^aC^
Chia + OL	838.1 ± 3.9 ^bC^	0.47 ± 0.03 ^aC^
Sesame	826.9 ± 4.9 ^aC^	3.12 ± 4.9 ^bB^
Sesame + OL	731.4 ± 10.9 ^bB^	5.39 ± 0.09 ^aA^
OL	535.2 ± 4.3 ^A^	-

Mean ± standard deviation of duplicates. The lower-case letters in the column indicate a significant difference between oil yields extracted with and without OL by the t-student mean test (*p* < 0.05). Different capital letters in the column indicate a significant difference between oil production of different seeds by Tukey test (*p* < 0.05). EC_50_: Ability to inhibit 50% of free radical DPPH. OL: olive leaves.

## Data Availability

Not applicable.

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
