# Peer review of "Simultaneous Extraction of Bioactive Compounds from Olea europaea L. Leaves and Healthy Seed Oils Using Pressurized Propane"

_foods, 2023, doi:10.3390/foods12050948_

Round 1

Reviewer 1 Report

Dear authors, 

The manuscript submitted here contains important data. However, some major revisions must be done to improve the submission. Please find below my comments and suggestions: 

1-Language must be revised carefully. Likewise, typos have to be corrected, 

2-Line numbering is important to refer to. Line 2nd (abstract): please change components to compounds,

3-Line 4th (abstract): please revised this expression (incomparable quality), 

4-5th line (abstract):  please change "aimed to" to "aimed at",

5-Please use abbreviations instead of full names (e.g, olive leaf). Full name at first mention only,

6-The state-of-art of research has to be deepened several recently published works can be included in the Introduction section: https://doi.org/10.3390/foods11203258;etc)

7-Page 3/12 (line 3): please add author name but without publication year,

8-Please check the values in Table 2,  

9-In Fig. 2, please change F (Factor: X and Y axis)  to PC (Principal Component),

10-I see that other multivariate methods can be used to better exploit the data: Cluster analysis, correlations, ANN, etc,

11-Why did you not measure the routine quality indices (for vegetable oils)?to evaluate effects of enrichment using OL: acid value, peroxide value, K232, K270, etc.,

12- The Results must be discussed in-depth 

Kind regards. 

Author Response

The manuscript submitted here contains important data. However, some major revisions must be done to improve the submission. Please find below my comments and suggestions: 

AU: Thank you. All changes were performed and are highlighted in yellow in the manuscript.

1-Language must be revised carefully. Likewise, typos have to be corrected, 

AU: Thank you for your revision. The manuscript was revised for English language by a professional and the certificate is amended to the submission.

2-Line numbering is important to refer to. Line 2nd (abstract): please change components to compounds,

AU: Thank you for your revision. Line numbers were inserted into the manuscript, and the requested change was made.

3-Line 4th (abstract): please revised this expression (incomparable quality), 

AU: Thank you for your revision. The expression was replaced.

4-5th line (abstract):  please change "aimed to" to "aimed at",

AU: Accepted. The change was made.

5-Please use abbreviations instead of full names (e.g, olive leaf). Full name at first mention only,

AU: Accepted. Abbreviations were corrected and standardized.

6-The state-of-art of research has to be deepened several recently published works can be included in the Introduction section: https://doi.org/10.3390/foods11203258;etc)

AU: Accepted. New studies were added in the introduction. Please, see L. 45-47, 80-84.

7-Page 3/12 (line 3): please add author name but without publication year.

AU: Thank you for your revision. The author's name without publication year was added. Please, see L. 107.

8-Please check the values in Table 2,  

AU: Thank you for your revision. Values were checked. Some values are equal due to rounding. Very low values were rounded for statistical analysis. Statistical analysis refers to seed oil with OL and without OL (chia/chia+OL) and (sesame/sesame+OL)

9-In Fig. 2, please change F (Factor: X and Y axis)  to PC (Principal Component),

AU: Thank you for your revision. The change was made.

10-I see that other multivariate methods can be used to better exploit the data: Cluster analysis, correlations, ANN, etc,

AU: Thank you for your comment. We tried to run an HCA for the parameters, but the effect of oil type was much higher than the effect of OL extract, and clustering was impossible. Furthermore, performing an HCA only with 2 samples is not recommended if we made the oils separately. We included the Pearson correlation and improved the discussion.

11-Why did you not measure the routine quality indices (for vegetable oils)? To evaluate effects of enrichment using OL: acid value, peroxide value, K232, K270, etc.,

AU: Thank you for your proposal. The objective of the article is to evaluate the bioactive compounds of the oil and the increase in oxidative stability and antioxidant activity of oils, according to previous studies (https://doi.org/10.1016/j.crfs.2022.03.002, https://doi .org/10.1016/j.supflu.2018.08.002, https://doi.org/10.1016/j.supflu.2019.104568,  https://doi.org/10.1016/j.supflu.2021.105189, but in the next studies we will carry out these analyzes to improve the quality of the work.

12- The Results must be discussed in-depth 

AU: Accepted. The results were discussed in-depth.  See L. 217-222, 253-260, 238-347, 360-373.

Kind regards. 

Reviewer 2 Report

The original paper presented by the authors has some scientific relevance. In this study, the authors propose the enrichment of vegetable oils, namely chia and sesame, with bioactive compounds obtained from the introduction of olive leaves in the extraction process using pressurized propane. Just from the relevance of the work, I believe that many points of the manuscript can be improved, such as clarity in writing, the structure of the text, and the discussion of results. Below I present some suggestions: “

Correct…synthetic antioxidants are carcinogenic…” to “..some synthetic antioxidants may have carcinogenic potential”  

The varieties of sesame and chia used in the experiments must be describe.

 The abbreviation (OL) indicated at the beginning of the of introduction must be used in the text. Sometimes it is used for olive leaves, other times for olive leave extract. This should be standardized.

Correct “described by [29]” to “described by Trentini et al. [29]”

  Add GC-MS details (brand, model…)

Correct “with BF3” to “with borontrifluoride (BF3)”

Add a reference to equation 1-4.

Add a reference to  the phytosrterols and tocopherols derivatization method.

 Complete the sentence : “…found 27% in chia (CH) and.”

Remove the double caption from figure 1

Correct “…not statistically different (p<0.05)” to “…not statistically different (p>0.05)”

 The authors can discuss the results of the DFA and SFA indexes

Table 3. Remove the letters where they are not necessary correct title of the table 5.

 Correct the sentence: ”Statistical analysis showed that sesame + OL was the oil with the highest antioxidant activity with EC50 731.4 ± 10.9 μg·mL-1, followed by CH + OL with EC50 914.1 ± 10.9 μg·mL-1”

 Correct: “Similar results were obtained by [25]”. Please, add author name after by.

 Correct “Table 4 shows the induction time” to “ Table 5 shows…”

add the meaning of the abbreviation IT

 According to the PCA, chia oils were characterized by their antioxidant activity, but sesame oils showed better activity (table 5). I suggest a transformation of the EC50 data (e.g., 1/EC50) after running the PCA.

Author Response

The original paper presented by the authors has some scientific relevance. In this study, the authors propose the enrichment of vegetable oils, namely chia and sesame, with bioactive compounds obtained from the introduction of olive leaves in the extraction process using pressurized propane. Just from the relevance of the work, I believe that many points of the manuscript can be improved, such as clarity in writing, the structure of the text, and the discussion of results. Below I present some suggestions: “

AU: Thank you. All changes were performed and are highlighted in yellow in the manuscript.

Correct…synthetic antioxidants are carcinogenic…” to “..some synthetic antioxidants may have carcinogenic potential”  

AU: Thank you for your revision. The change was made. Please, see L. 37-38.

The varieties of sesame and chia used in the experiments must be describe. 

AU: Accepted. Chia and sesame varieties were mentioned. Please, see L. 94-95. 

 The abbreviation (OL) indicated at the beginning of the of introduction must be used in the text. Sometimes it is used for olive leaves, other times for olive leave extract. This should be standardized.

AU: Accepted. The abbreviation "OL" was used only for "olive leaves", when mentioning olive leaves extract it was described as "OL extract".

Correct “described by [29]” to “described by Trentini et al. [29]”

AU: Thank you for your revision. The change was made. Please, see L. 107.

Add GC-MS details (brand, model…)

AU: Accepted. Details were added.

Correct “with BF3” to “with borontrifluoride (BF3)”.

AU: Thank you for your revision. The change was made. Please, see L. 129.

Add a reference to equation 1-4. 

AU: Accepted. References were added. Please, see L. 143.

Add a reference to the phytosrterols and tocopherols derivatization method. 

AU: Accepted. The methodology was described in more detail and references were added. See L. 128-139. 

Complete the sentence : “…found 27% in chia (CH) and.”

AU: Thank you for your revision. The sentence was misspelled, so the word "and" was dropped.

Remove the double caption from figure 1

AU: Thank you for your revision. The double caption was removed.

Correct “…not statistically different (p<0.05)” to “…not statistically different (p>0.05)”

AU: Thank you for your revision. The sentence was modified.

The authors can discuss the results of the DFA and SFA indexes

AU: Accepted. The discussion was inserted. See L. 253-260.

Table 3. Remove the letters where they are not necessary correct title of the table 5.

AU: Thank you for your revision. Unnecessary letters were removed. The title of Table 5 has been rewritten.

Correct the sentence: ”Statistical analysis showed that sesame + OL was the oil with the highest antioxidant activity with EC50 731.4 ± 10.9 μg·mL-1, followed by CH + OL with EC50 914.1 ± 10.9 μg·mL-1”.

AU: Accepted. The sentence was rewritten. Please, see L. 332-334. 

 Correct: “Similar results were obtained by [25]”. Please, add author name after by. 

AU: Thank you for your revision. Correction was made. Please, see L. 332.

 Correct “Table 4 shows the induction time” to “Table 5 shows…”

AU: Thank you for your revision. Correction was made.

add the meaning of the abbreviation IT

AU: Thank you for your revision. The meaning of the abbreviation was added. Please, see L. 350.

 According to the PCA, chia oils were characterized by their antioxidant activity, but sesame oils showed better activity (table 5). I suggest a transformation of the EC50 data (e.g., 1/EC50) after running the PCA.

AU: Thank you for your revision. The highest EC50 value (ability to inhibit 50% of the DPPH free radical) indicates lower antioxidant activity, as mentioned in L. 325-326. Therefore, there was a mistake in saying that it was the highest antioxidant activity in Chia Oil the value is higher, but it does not indicate greater antioxidant activity.

Round 2

Reviewer 1 Report

Dear Authors, 

Thank you for taking my comments into account to improve the manuscript. I see that the manuscript can be accepted for publication in Foods. 

Kind regards. 

Reviewer 2 Report

The authors made all corrections in the manuscript indicated by the reviewers. I believe that there was an improvement in the article and that it can be accepted for publication.

Greetings and good luck